# Drug-Target Network Study Reveals the Core Target-Protein Interactions of Various COVID-19 Treatments

**DOI:** 10.3390/genes13071210

**Published:** 2022-07-06

**Authors:** Yulin Dai, Hui Yu, Qiheng Yan, Bingrui Li, Andi Liu, Wendao Liu, Xiaoqian Jiang, Yejin Kim, Yan Guo, Zhongming Zhao

**Affiliations:** 1Center for Precision Health, School of Biomedical Informatics, The University of Texas Health Science Center at Houston, Houston, TX 77030, USA; yulin.dai@uth.tmc.edu (Y.D.); qiheng.yan@uth.tmc.edu (Q.Y.); bingrui.li@uth.tmc.edu (B.L.); andi.liu@uth.tmc.edu (A.L.); wendao.liu@uth.tmc.edu (W.L.); 2Comprehensive Cancer Center, Department of Internal Medicine, The University of New Mexico, Albuquerque, NM 87131, USA; huiyu1@salud.unm.edu; 3Department of Biostatistics and Data Science, School of Public Health, The University of Texas Health Science Center at Houston, Houston, TX 77030, USA; 4Metastasis Research Center, Department of Cancer Biology, The University of Texas MD Anderson Cancer Center, Houston, TX 77030, USA; 5Department of Epidemiology, Human Genetics and Environmental Sciences, School of Public Health, The University of Texas Health Science Center at Houston, Houston, TX 77030, USA; 6MD Anderson Cancer Center UTHealth Graduate School of Biomedical Sciences, Houston, TX 77030, USA; 7Center for Secure Artificial Intelligence for Healthcare, School of Biomedical Informatics, The University of Texas Health Science Center at Houston, Houston, TX 77030, USA; xiaoqian.jiang@uth.tmc.edu (X.J.); yejin.kim@uth.tmc.edu (Y.K.); 8Human Genetics Center, School of Public Health, The University of Texas Health Science Center at Houston, 7000 Fannin St. Suite 600, Houston, TX 77030, USA

**Keywords:** COVID-19, text mining, drug-target network, drug treatment

## Abstract

The coronavirus disease 2019 (COVID-19) pandemic has caused a dramatic loss of human life and devastated the worldwide economy. Numerous efforts have been made to mitigate COVID-19 symptoms and reduce the death rate. We conducted literature mining of more than 250 thousand published works and curated the 174 most widely used COVID-19 medications. Overlaid with the human protein–protein interaction (PPI) network, we used Steiner tree analysis to extract a core subnetwork that grew from the pharmacological targets of ten credible drugs ascertained by the CTD database. The resultant core subnetwork consisted of 34 interconnected genes, which were associated with 36 drugs. Immune cell membrane receptors, the downstream cellular signaling cascade, and severe COVID-19 symptom risk were significantly enriched for the core subnetwork genes. The lung mast cell was most enriched for the target genes among 1355 human tissue-cell types. Human bronchoalveolar lavage fluid COVID-19 single-cell RNA-Seq data highlighted the fact that T cells and macrophages have the most overlapping genes from the core subnetwork. Overall, we constructed an actionable human target-protein module that mainly involved anti-inflammatory/antiviral entry functions and highly overlapped with COVID-19-severity-related genes. Our findings could serve as a knowledge base for guiding drug discovery or drug repurposing to confront the fast-evolving SARS-CoV-2 virus and other severe infectious diseases.

## 1. Introduction

Coronavirus disease 2019 (COVID-19) has spread globally, with over 422 million confirmed cases and over 5.8 million deaths worldwide as of 20 February 2022 [1]. SARS-CoV-2 belongs to the coronavirus (CoV) family, which includes life-threatening respiratory diseases such as severe acute respiratory syndrome (SARS) and Middle East respiratory syndrome (MERS) that typically spread from animal hosts such as bats and civet to humans [2]. During the past two years, medical personnel and investigators around the world have spared no effort to explore medical treatments and develop potential vaccines for SARS-CoV-2 [3]. This research provides us with critical information underlying massive amounts of heterogeneous data, especially potential drugs with high efficacy for treating COVID-19 and the associations among drugs, genes, and the coronavirus. Drugs involved in COVID-19 clinical trials belong to four major categories in the classification in PharmGKB [4], including those that inhibit viral entry, those that inhibit viral replication, anti-cytokine (anti-inflammatory) drugs, and others. Among them, a few antiviral drugs such as Remdesivir [5], Paxlovid [6], and Molnupiravir [7], and multiple monoclonal antibodies [8] have been approved by the FDA. Drugs suppressing viral replication mainly target the SARS-CoV2 polymerase and the replication process [9]. Therefore, they do not directly target the human cellular interactome. Drugs that inhibit viral entry were designed to block the interaction between the spike protein of SARS-CoV-2 and the human cell surface protein ACE2 and transmembrane protease TMPRSS2 [10]. Drugs in the anti-cytokine category are intended to mitigate the severe COVID-19 symptoms induced by a hyperinflammatory immune response [11]. As well as these, various other drugs have been identified as conditionally effective treatments for COVID-19 and related symptoms [12,13,14]. However, the fast evolution of the virus has threatened the efficacy of both vaccines and drug treatments [15,16]. Understanding their pharmacological process will help us to measure the usability of individual drugs and drug combinations and eventually benefit from the discovery of potential treatments for other infectious diseases.

Systematic identification of drug-target and target-protein interactions can effectively explain the underlying mechanisms of drugs [17,18,19]. Network pharmacology methods have been used to transform drug discovery technology from developing single target ligands to more clinically effective drugs that target multiple proteins [20,21]. The applications of these concepts in drug research include target identification, target-protein interaction analysis, side-effect prediction, and molecular transport analysis [22]. The network analysis involved in biomedical research helps researchers and physicians to understand the mechanisms of drugs and prioritize the treatments for patients [23,24]. Viruses typically require host cellular factors in order to successfully enter the cells and replicate during infection [25]. After the viral particles enter the host cell, the host innate immune response is initiated via the production of type I interferons (IFN-α/β), activation of the JAK-STAT pathway [26], and the subsequent recruitment of a series of pro-inflammatory cytokines [27]. The dysregulation of these pro-inflammatory responses leads to severe COVID-19 symptoms, including fever, cytokine storm, and acute respiratory distress syndrome [28]. Systematic analysis of the known drugs that are involved in virus–host interaction and the host immune response regulatory network will guide us in understanding effective strategies for combating COVID-19 and for drug repurposing [29,30].

In this study, we aimed to understand COVID-19 drug targets on the human cellular network and their relevance to COVID-19-related disease genes. (1) We conducted systematic curation of COVID-19-related treatments and their corresponding targets. (2) We projected them into a human protein–protein interaction (PPI) reference and used the Steiner tree to connect the core target genes into an actionable network. (3) To explore the features of the network, we conducted functional enrichment analysis and cell-type-specific enrichment analysis. (4) We further explored the drug target–protein network with COVID-19-related genes from GWAS risk genes and single-cell RNA-seq data.

## 2. Materials and Methods

### 2.1. Text Mining of COVID-19 Drugs and Drug Target Curation

Literature was firstly collected from PubMed (https://pubmed.ncbi.nlm.nih.gov, accessed on 4 October 2021), using the keyword set [“COVID-19”, “COVID 19”, “SARS-CoV-2”, “SARS COV 2”]. All abstracts with matching keywords were downloaded from PubMed using the “batch_pubmed_dowload” function in the easyPubMed R package (https://cran.r-project.org/web/packages/easyPubMed/index.html). The easyPubmed package is an R interface allowing easy programmatic access to PubMed. Then, we adopted a natural language processing (NLP) model, Med7, to extract all drug names from the downloaded abstracts [31]. Med7 is a transferable clinical NLP model for electronic health records. It has been trained to recognize seven categories from electronic health record (EHR) data: the amount of drug administered, the name of the drug, the length of prescription, the form of drug given, the dosage regimen of the drug, the route for the drug to enter the body, and the amount of drug in each dose. It is extremely useful for extracting drug-related information from EHR data or texts in general [31]. Roughly the top 20% of the drugs were selected, based on the distribution of the number of occurrences of the extracted drugs (log10(freq) > 1), with subsequent filtration of noisy drug names outputted from Med7. We further manually inspected the abstracts for the remaining drugs. Drugs that were not directly used or proposed for treating COVID-19 in other literature studies were filtered out from our list. Finally, target-gene information was collected from DrugBank using the “dbparser” R package (https://cran.r-project.org/web/packages/dbparser/index.html) on 12 January 2022 [32].

### 2.2. Steiner Tree Analysis

Steiner tree [33] is a subnetwork extraction algorithm that identifies the least number of mediator nodes required to interconnect the input terminal nodes. The algorithm has been applied in systems biology research [34,35]. We obtained all non-redundant protein–protein interactions from BioGRID (version 4.4.203) [36], including 19,094 genes and 539,890 unique interactions after removing non-human and redundant data. We derived a COVID-19-related parental network from these interactions by restricting the search within our curated COVID-19 drug targets. We identified the common drugs between our curation and the COVID-19 drug curation from the Comparative Toxicogenomics Database (CTDbase) [37]. Then, we took the union targets of the common drugs as the terminal nodes, to carry out the Steiner tree analysis in the COVID-19-related parental network. The Steiner tree algorithm iteratively added the next mediator node(s) with the minimal average shortest path to the existing isolated tree components until the isolated components were merged into one single component. In the resultant interconnected subnetwork, all input genes appeared as terminal nodes, whereas the algorithm-selected additional nodes were placed at the inner parts as mediators. Only a minimum number of interactions were preserved in the subnetwork to interconnect all terminals and mediators. The identified mediator nodes were considered topologically important because they were the optimal set of nodes bridging the terminal nodes.

We investigated which drugs were enriched in the Steiner tree subnetwork resulting from the parental network. For each drug with at least one retained target gene, the target genes were counted into both the Steiner tree subnetwork and the parent subnetwork. The target gene reservation rate was compared with the subnetwork size shrinkage rate through a hypergeometric test, and drugs with a *p*-value of less than 0.01 were considered to be plausible COVID-19 drugs.

### 2.3. Functional Enrichment Analysis

To investigate the features of target genes, we performed an over-representation analysis using the R package WebGestaltR (https://cran.r-project.org/web/packages/WebGestaltR/index.html, accessed on 10 February 2022) with no redundant Gene Ontology (GO) terms (Biological Process, Molecular Function, and Cellular Component), with all human protein-coding genes as the reference. We used Benjamini–Hochberg (BH) approach to adjust the *p*-value [38]. To understand the cell-type-specific enrichment analyses (CSEA) of the target genes, we input the genes from the core target interaction network to our in-house tool, Web-based Cell-type-Specific Enrichment Analysis (WebCSEA, https://bioinfo.uth.edu/webcsea/, accessed on 15 February 2022) [39,40]. Specifically, this online tool utilized our previous deTS algorithm [41] to calculate the raw *p*-value across 1355 tissue-cell types. To overcome the potential bias due to the different lengths of signature genes among tissue-cell types, we calculated the permutation *p*-value by ranking the raw *p*-value with >20,000 gene lists from GWAS and a rare-variants association of human traits and disease pre-curated in WebCSEA [42,43]. Overall, we calculated a combined *p*-value with two significant thresholds to evaluate the significance. The suggestive significance was 0.001. The stringent significance was defined as a Bonferroni-corrected *p*-value of 3.7 × 10^−5^ (0.05/1355).

### 2.4. GWAS Summary Statistics Process and z-Score Permutation

We collected six COVID-19 European ancestry GWAS summary statistics available on 23 August 2020. The related COVID-19 GWAS included the following traits: “Severe COVID-19 infection with respiratory failure (analysis I) and (analysis II)” from the severe COVID-19 GWAS group [44], “hospitalized COVID-19 vs. not hospitalized COVID-19”, “predicted COVID-19 self-reported symptoms vs. predicted or self-reported non-COVID-19” from the COVID-19 Host Genetics Initiative [45] and the UK biobank COVID-19 study “COVID-19 UKBB tested controls”, and “COVID-19 UKBB tested controls” from the Genome-Wide Repository of Associations Between SNPs and Phenotypes (GRASP) [46]. The first three traits are COVID-19-severity-related phenotypes, while the other three traits are COVID-19-susceptibility-related phenotypes. The detailed GWAS description is available in the Appendix A. We used the multi-marker analysis of GenoMic Annotation (MAGMA v1.07) to calculate the gene-level *p*-value [47]. MAGMA combines multiple SNPs mapped to the same gene and adjusts the effects of the gene length, SNP density, and local linkage disequilibrium (LD) structure. We considered all SNPs located in the window from 50 kb upstream to 35 kb downstream. We used the mean of the χ2 statistic for the SNPs to measure the gene-level *p*-value for each gene. We used the 1000 Genome Project Phase 3 European population as the reference panel.

We adapted the gene-level z-score transformed from the MAGMA output, which was calculated from the inversed probit function Φ:
(1)$$Z_{i}=\Phi^{-1}\left(1-P_{i}\right),$$

Here, *Pi* is the gene-level *p*-value. We calculated the mean of the z-scores of the focal gene list. Then, we randomly selected the same number of genes as the focal gene list from the whole GWAS gene set without replacement one million times, to obtain one million medium z-scores from permutation. Lastly, we defined the permuted *p*-value as the proportion of cases from one million permutations that returned a z-score higher than the focal gene list z-score.

### 2.5. Differentially Expressed Gene Analysis for COVID-19 Single-Cell RNA-Seq Data

We obtained the COVID-19 BALF single-cell RNA sequencing (scRNA-seq) data from 13 patients (severe (n = 6), moderate (n = 3), and healthy (n = 4)) generated by Liao et al. (GSE145926) [48]. The processed data, with disease severity and cell-type annotation from the original study, were downloaded from https://covid19-balf.cells.ucsc.edu (accessed on 11 October 2021) and used in our analysis. We compared all the differentially expressed genes (DEGs) between the severe group and the healthy group, as well as between the severe and moderate groups, across B cells, epithelial cells, macrophages, myeloid dendritic cells (mDCs), neutrophils, NK cells, plasmacytoid dendritic cells (pDCs), plasma cells, and T cells. We performed a non-parametric Wilcoxon rank sum test for differential expression analysis, using the “FindMarkers” function in the “Seurat” R package [49].

## 3. Results

### 3.1. Identifying COVID-19 Drugs and Corresponding Targets via Literature Mining and Curation

We developed a standard literature mining workflow to obtain COVID-19-related drugs and their corresponding human target genes (Figure 1). Specifically, we searched PubMed using the keyword set [“COVID-19”, “COVID 19”, “SARS-CoV-2”, “SARS CoV 2”], which led to > 250,000 non-redundant abstracts. Then, we used a clinical natural language tool, Med7, to extract all drug names from the downloaded abstracts. In total, 1419 drug names were detected and collected by Med7. To study the most widely used drugs and filter out potential noise, we set an empirical cutoff of log10 (times drug mentioned in the collected abstracts) > 1 to prioritize the drug list. Therefore, roughly the top 20% (269) of the drugs were left (Appendix A). Next, we manually curated the abstracts containing these 269 candidates to exclude drugs with a potentially negative effect on COVID-19 outcomes. As a result, 212 drugs that were directly used or proposed for treating COVID-19 were retained for the drug target curation. To avoid the inconsistent annotation strategies among different drug annotation databases [50], we collected the target gene information from DrugBank [51]. Overall, we curated 803 unique genes targeted by 174 distinct drugs (Appendix A). We also mapped these drugs to the PharmGKB database and identified 12, 7, and 16 drugs that were annotated for inhibiting viral entry, inhibiting viral replication, and anti-cytokine/anti-inflammatory function, respectively (Appendix A and Appendix A).

### 3.2. COVID-19 Drug Target-Protein Network

We constructed a global PPI reference network using experimentally validated data from BioGRID [52], in which we identified 783 genes out of 803 unique genes from our compiled COVID-19 drug targets. In accordance with one comprehensive viral–host PPI study [53], we identified another set of 314 “host factor” proteins in the PPI network that were confirmed to have a physical interaction with the SARS-CoV-2 virus in human cell lines. The shortest paths between our drug target genes and the host factors in the global BioGRID network averaged 2.46, whereas the shortest path distance between any two genes in the global network was 2.86. We randomly sampled 314 vertices from the global network and recorded the average shortest path distance between our drug target genes and the random vertex set. For 100 random sampling experiments, the average shortest path distance was always larger than the observed distance of 2.46 between our drug target genes and the 314 experimentally confirmed host factors (Appendix A). Therefore, our compiled 783 COVID-19 drug target genes were significantly more adjacent to SARS-CoV-2 host factors than random vertices in the BioGRID PPI network (*p* < 0.01).

Next, we extracted a medium-scaled subnetwork from the global PPI network that only involved our curated COVID-19 drug targets. This subnetwork excluded the isolated genes that were not directly connected to the major component. It consisted of 4245 edges of 680 genes (Appendix A). We denoted this medium-scaled subnetwork the COVID-19-related parental network, because we intended to next narrow it down to a small-scaled subnetwork. The vertex degree and vertex frequency showed a linear relationship on a logarithm scale (Appendix A), conforming to the typical scale-free property of a molecular biology network. The degree and betweenness of vertices had a Pearson correlation coefficient of 0.84. Genes of the highest degree included the well-known transcription factor genes TP53, MYC, and EGFR (Table 1).

Then, we cross-validated our parental network with another COVID-19 drug curation from CTDbase [37] (data accessed on 4 January 2022). There were 10 common COVID-19-related drugs between CTDbase and our compilation (Table 2). The union of these target genes had 25 vertices overlapped with the COVID-19-related parental network, which were not fully interconnected. Lastly, we took these 25 genes as credible COVID-19 drug targets and employed the Steiner tree algorithm [34] to extract a subnetwork from the parental PPI network that most parsimoniously interconnected the 25 terminal genes. Finally, this resulted in a COVID-19-related core subnetwork (COVID19-DrugNET) involving 34 genes (25 terminals and 9 mediators) with 47 edges (Figure 2A, Appendix A). In addition to the 10 CTDbase-curated drugs, another 26 drugs were associated with these 34 core drug target genes, and 10 of the 26 drugs had their target genes significantly enriched in COVID19-DrugNET by the COVID-19-related parental network (Table 3).

### 3.3. Functional Enrichment and Tissue-Cell-Type Specificity of COVID19-DrugNET Genes

As shown in Figure 2B, the over-representation analysis of COVID19-DrugNET genes highlights the following functions: (1) cell surface receptor responses to an external stimulus such as “GO:0032103, positive regulation of response to external stimulus (P_BH_ = 4.34 × 10^−13^)”; “GO:0005126, cytokine receptor binding (P_BH_ = 3.62 × 10^−12^)”; “GO:0097696, STAT cascade (P_BH_ = 1.59 × 10^−10^)”; “GO:0001664, G protein-coupled receptor binding (P_BH_ = 3.79 × 10^−7^)”and (2) immune responses such as “GO:0002526, acute inflammatory response (P_BH_ = 1.91 × 10^−7^)”; “GO:0002237, response to molecule of bacterial origin (P_BH_ = 1.91 × 10^−7^)”; and ”GO:0050727, regulation of inflammatory response (P_BH_ = 3.54 × 10^−7^)”.

We conducted a cell-type-specific enrichment analysis (CSEA), using our in-house method, for the 34 COVID19-DrugNET genes (Figure 3A,B) [42,54] and identified that lung mast cell has a nominal significance (P_adjust_ = 0.0003, Figure 3C). The mast cell is a long-lived tissue-resident cell with an important role in immune response, indicating that our COVID19-DrugNET genes could be targeted to human lung mast cells that mitigate severe COVID-19 symptoms [55,56]. In addition, microglia in the fetal cerebellum (P_adjust_ = 0.001) and monocyte in the adult liver (P_adjust_ = 0.001) also reached nominal significance. Overall, immune-related cell types were mostly enriched by COVID19-DrugNET.

### 3.4. COVID19-DrugNET Is Highly Related to Risk Genes Underlying Severe COVID-19 Symptoms

Genetic factors play important roles in terms of COVID-19 severity and susceptibility [44,57,58]. To test whether our COVID19-DrugNET genes had an average higher risk for severe COVID-19 symptoms, we further explored the 34 genes from the core network relevant to COVID-19 GWAS traits. Specifically, we obtained summary statistical data for six GWAS, for COVID-19-related traits from case-control studies (Appendix A) and performed a GWAS z-score permutation test for the genes from our COVID19-DrugNET. We identified that our gene list had significantly higher mean z-score enrichments for one of the COVID-19 severe symptoms traits, i.e., “Severe COVID-19 infection with respiratory failure (analysis I)”, (*p* = 0.049, Appendix A). We failed to identify any significant *p*-value from three COVID-19 susceptibility-related traits, suggesting our COVID19-DrugNET genes overall had severity-related risks rather than susceptibility-related risks.

To analyze the relationships between COVID19-DrugNET genes and COVID-19-disease-related products, we adapted one COVID-19 scRNA-seq dataset from bronchoalveolar lavage fluid (BALF) [48]. We systematically compared the COVID19-DrugNET genes with DEGs between the COVID-19 severe vs. the healthy group (Figure 4A) and the severe vs. the moderate group (Figure 4B) (Appendix A). We identified the corresponding number of overlapping genes in macrophages (16:14) and T cells (14:11) among the comparison groups (Figure 4A,B). These findings indicate that the COVID19-DrugNET genes overlapped with almost half of the COVID-19 dysregulated genes in T cells and macrophages of severe disease patients. These are the major contributors of cytokine storm and hyperinflammatory response [59].

## 4. Discussion

This work explored the COVID-19 drugs from >250 K literature studies using Med 7 and validated them with other resource curations. First, through a series of filtrations, we identified ten drugs shared with the drugs list from CTDbase. We used Steiner tree analysis to connect the target genes of the ten drugs from the human reference PPI, which led to the final COVID19-DrugNET, containing 34 genes and 47 edges. These genes are enriched with “response to external stimulus” and “immune response” functions. Interestingly, the tissue and cell-type enrichment analysis for the 34 genes identified that the lung mast cell (the resident immune system in the lung) had the most significant signal. Lastly, the COVID-19 scRNA-seq DEGs analysis and severe GWAS phenotypes all indicated that our COVID19-DrugNET genes highly overlapped with COVID-19-severity-related genes.

Our findings on the overlapping ten drugs with human gene targets are mainly related to anti-cytokine/anti-inflammatory and other unknown drug mechanisms, according to the COVID-19 drug classification of PharmGKB (Table 2). The drug baricitinib is a Janus kinase (JAK) inhibitor for treating adult patients with moderate-to-severe active rheumatoid arthritis via interfering with the pathway that leads to inflammation [60]. The drug tofacitinib is also in the JAK inhibitor class of drugs that suppress pro-inflammatory cytokine activity [61]. For the drugs in other categories: (1) acalabrutinib is a Bruton’s tyrosine kinase (BTK) inhibitor on the B-cell receptor signaling pathway that communicates with other immune cells and results in B-cell proliferation and activation [62] and (2) dapagliflozin is a sodium–glucose cotransporter 2 inhibitor used as the antihyperglycemic treatment for diabetes. This drug showed clinical status improvement, although it was not statistically significant, in a phase 3 trial in patients with cardiometabolic risk factors [63]. Finally, (3) ruxolitinib is a JAK inhibitor with a similar mechanism to baricitinib [64].

For the other five drugs without PharmGKB annotation: (1) The non-peptidic drug aliskiren could interact with the catalytic site of SARS-CoV-2 main protease and interfere with the viral function [65]; (2) argatroban is a trypsin-like serine protease, which has potential therapeutic benefits in COVID-19 patients via its antithrombotic, anti-inflammatory, and antiviral effects [66]; and (3) bicalutamide is an antiandrogen therapy for male prostate cancer, which was used to tackle the viral entry via regulating TMPRSS2. However, one recent phase 2 clinical trial failed to identify significant improvement from using this drug [67]. (4) The drug ibrutinib is a kinase inhibitor that decreases the B-cell proliferation and survival by irreversibly blocking the BTK B-cell receptor pathway and was reported to treat COVID-19 hyperinflammation [68]; and (5) montelukast is a cysteinyl leukotriene receptor antagonist with an anti-inflammatory effect, cytokine production reduction, and oxidative stress suppression [69].

Overall, eight out of these ten drugs are either in phase 2/3 or phase 4 of clinical trials (Table 2). Their targets and involved pathways are mainly related to anti-cytokine activity and inhibiting viral entry, which explains the functions of the module genes enriched in the immune cell membrane receptors and the downstream cellular signaling cascade. Nevertheless, many antiviral agents and anti-inflammatory drugs have been reported for combination use [70,71,72], providing better treatment efficacy than monotherapies.

We conducted a systematic exploration to understand the features of the 34 COVID19-DrugNET genes derived from 36 credible drugs. Our cell-type-specific enrichment analysis identified that the resident innate immune cell (mast cell) in the lung, microglia in the fetal cerebellum, and monocyte in the adult liver are enriched with COVID19-DrugNET genes with a nominal significance, indicating that immune-related cell types are the major cellular targets of COVID19-DrugNET genes. Moreover, our GWAS z-score permutation identified that the COVID19-DrugNET genes have higher mean z-scores than random gene sets in GWAS severity-related traits, not in susceptibility-related traits. These findings all align with the composition of 34 COVID19-DrugNET genes, which are mainly anti-cytokine/anti-inflammatory drug targets for treating patients with severe symptoms. Our, scRNA-seq analysis of BALF COVID-19 data suggests that the macrophages and T cells contain more COVID19-DrugNET drug targets for treating severe COVID-19 patients, probably raised by the hyperinflammation and cytokine storms [59].

Lastly, this is a fast-moving field. Our approach might not capture all the latest drugs. After we finished our literature-mining on 4 October 2021, several new drugs have been approved by the FDA, including Paxlovid [6] and molnupiravir [7]. Although the antiviral replication drugs are the most effective monotherapies for mitigating virus activities directly and therefore reducing mortality rates, severe symptoms rates, and time to recovery, we expect to see more combination use of antiviral agents and anti-inflammatory drugs, which will shed new light on fighting SARS-CoV-2 infection.

## 5. Conclusions

We identified 174 COVID-19 drugs via extensive literature mining, including ten drugs shared with the CTDbase curation. We connected the targets of these ten drugs with PPI references and expanded them to a network module containing 34 genes that are enriched with membrane receptors of immune-related cell types and the downstream cellular signaling cascade. Our CSEA identified lung mast cell as the most relevant cell for COVID19-DrugNET. Genes in COVID19-DrugNET had higher than random GWAS z-scores, probably carrying severity-related rather than susceptibility-related genetic risks. Lastly, the DEGs of macrophages and T cells between severe and moderate/healthy individuals covered half of the drug targets from COVID19-DrugNET, indicating that these two cell types are the major targets of anti-inflammatory treatment for severe COVID-19 symptoms. Overall, our work constructed the COVID19-DrugNET, with drugs and therapeutic targets with high confidence, providing a systematic view of the underlying biological bases of various treatments.

## Figures and Tables

**Figure 1 genes-13-01210-f001:**
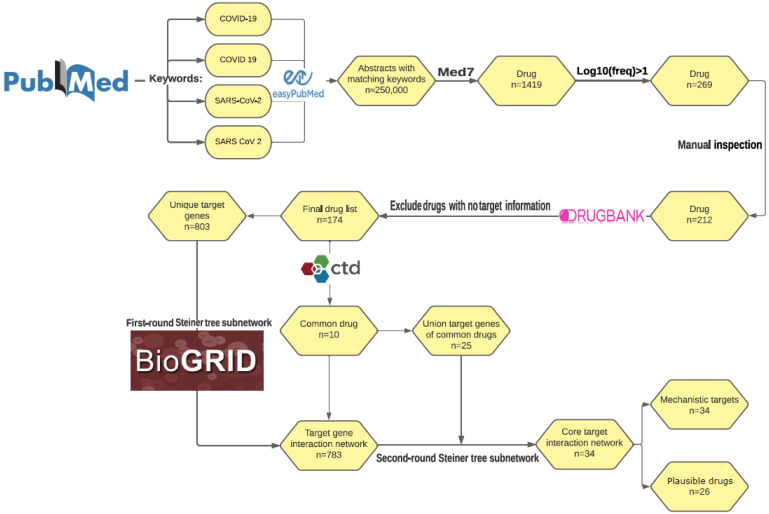
Literature mining and subnetwork extraction workflow: text mining of COVID-19 drugs, drug target curation, and Steiner tree network analysis. Abstracts with matching keywords were downloaded from PubMed. Drug names were extracted from the downloaded abstracts using Med7 and a cutoff was applied based on the empirical distribution, to further narrow down the drug list. Target gene information for each drug was collected from DrugBank. Starting from the target genes of ten credible drugs ascertained by CTDbase, Steiner tree algorithm was applied to a human protein–protein interaction network to extract a core target interaction subnetwork.

**Figure 2 genes-13-01210-f002:**
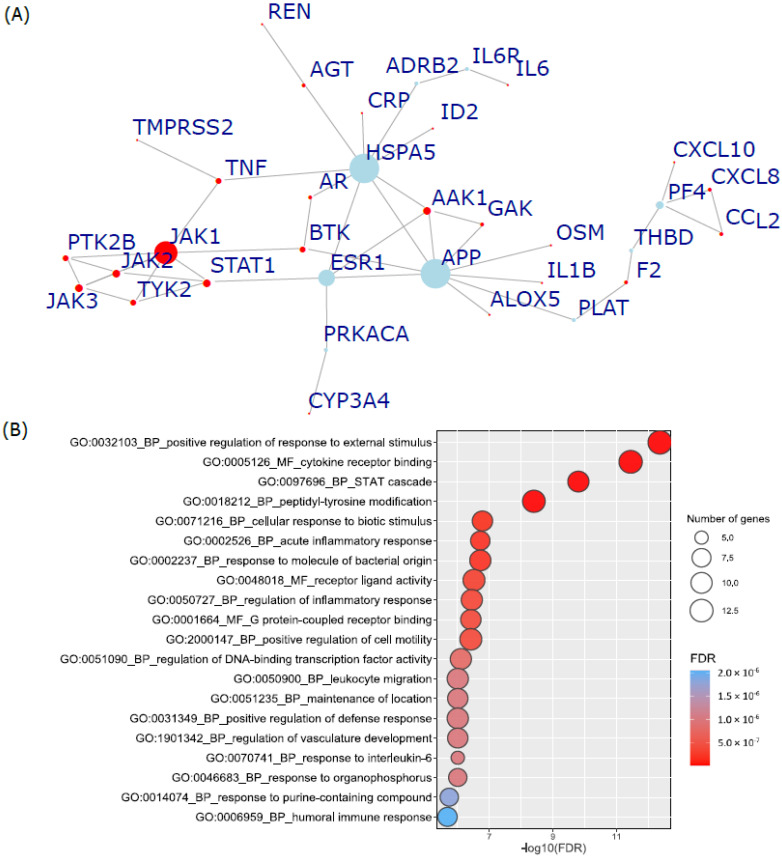
Steiner-tree-inferred protein–protein interaction network that interconnects 25 convincing COVID-19 drug target genes and functional enrichment. (**A**) Red node: credible COVID-19 drug target genes as terminals of the inferred Steiner tree. Blue node: a minimum set of genes (mediators) through which the interconnected subnetwork was formed. Node size was proportional to the degree. (**B**) Top 20 significant enrichment results for Gene Ontology (GO) analysis of biological process, molecular function, and cellular component. Each row is the GO term. The color of the circle is proportional to the value of −log10 (P_BH_) for each term, from blue to red. The circle size is proportional to the number of intersected genes between the 34 COVID19-DrugNET genes and the term genes.

**Figure 3 genes-13-01210-f003:**
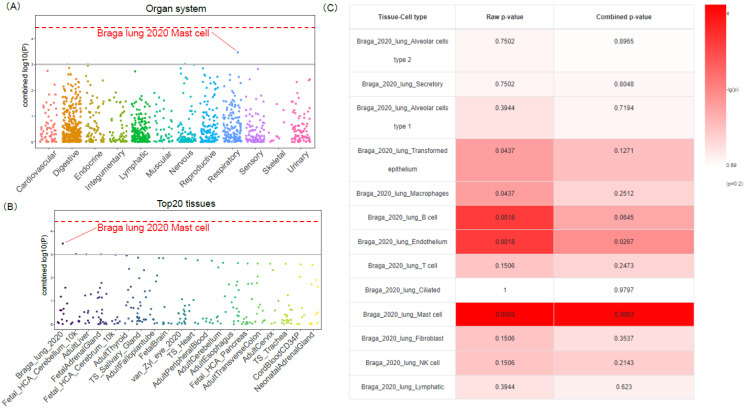
Cell-type-specificity of COVID19-DrugNET genes. The red dashed line indicates the Bonferroni-corrected significant threshold −log10 (*p* = 3.69 × 10^−5^). The grey solid line indicates the nominal significance −log10 (*p* = 1 × 10^−3^). (**A**) In each category of organ systems, each dot represents one tissue cell type from that organ system, in a different color by column. We highlighted the top cell type, i.e., lung mast cell in respiratory system. (**B**) In each category of tissue, each dot represents one cell type from that tissue, in a different color by column. We highlighted the top cell type, i.e., lung mast cell in one lung single-cell RNA sequencing (scRNA-seq) study. (**C**) Heatmap for the COVID19-DrugNET gene cell-type-specific enrichment analysis results in one lung scRNA-seq panel. The color is proportional to the *p*-values. The first column is the tissue cell type in this scRNA-seq panel. The second column is the raw *p*-values. The third column is the combined *p*-value calculated by WebCSEA.

**Figure 4 genes-13-01210-f004:**
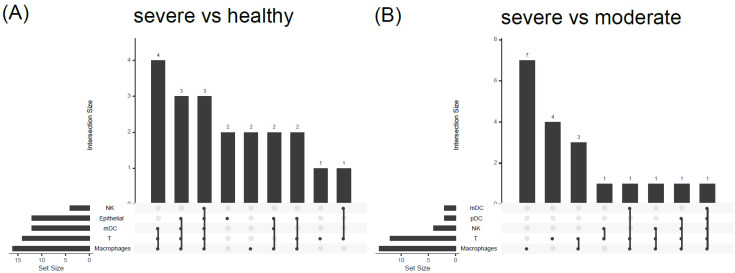
Comparison of COVID19-DrugNET genes with the differentially expressed genes (DEGs) from single-cell RNA-seq COVID-19 bronchoalveolar lavage fluid dataset. (**A**) In the severe and healthy groups, the UpSet plot shows the shared and uniqued components among the top 5 cell types with overlapping genes between the corresponding DEGs and COVID19-DrugNET genes. The set size indicates the overlapping genes. NK: natural killer cell; mDC: myeloid dendritic cell. (**B**) In the severe and moderate groups, the UpSet plot shows the shared and unique components among the top 5 cell types with overlapping genes between the corresponding DEGs and COVID19-DrugNET genes. The set size indicates the overlapping genes. mDC: myeloid dendritic cells; pDC: plasmacytoid dendritic cells.

**Table 1 genes-13-01210-t001:** The top ten genes with the highest degree in the COVID-19-related parental PPI network.

Gene Symbol	Degree	Betweenness
HSP90AA1	196	37,757.90
TP53	148	17,451.41
APP	145	32,447.83
NTRK1	141	15,363.83
MYC	134	12,058.01
EGFR	125	19,208.06
ESR1	114	7602.044
ESR2	89	8696.631
EGLN3	83	5614.034
XPO1	82	4984.01

**Table 2 genes-13-01210-t002:** The number of target genes of 10 common COVID-19-related drugs between our compilation and CTDbase.

Drug	Compilation	CTDbase	Overlapping	Union	PharmGKB Annotation
acalabrutinib ^†^	1	3	0	4	other
aliskiren	1	3	0	4	NA
Argatroban ^‡^	1	1	0	2	NA
baricitinib ^†‡^	4	11	3	12	anti-cytokine/anti-inflammatory
bicalutamide	1	4	0	5	NA
dapagliflozin ^†‡^	1	3	0	4	other
Ibrutinib ^†^	1	4	1	4	NA
montelukast ^†‡^	2	6	0	8	NA
ruxolitinib ^†^	4	4	0	8	other
tofacitinib ^†‡^	4	8	0	12	anti-cytokine/anti-inflammatory

We queried the clinical trials information of the 10 drugs on clinicaltrials.gov as of 28 February 2022. ^†^ Indicates that the drug was in phase 2/3 of clinical trial(s) for testing to treat COVID-19. ^‡^ Indicates that the drug was in phase 4 of clinical trial(s) for testing to treat COVID-19.

**Table 3 genes-13-01210-t003:** Plausible COVID-19 drugs with target genes significantly enriched in COVID19-DrugNET.

Drug	p.hyper	Target Reservation Rate	Reserved Target Genes
adalimumab	0	1/1	TNF
bromhexine	0.0018	1/2	TMPRSS2
canakinumab	0	1/1	IL1B
deferoxamine	0	1/1	APP
epinephrine	0.0036	2/8	ADRB2, TNF
formoterol	0.0053	1/3	ADRB2
infliximab	0	1/1	TNF
leflunomide	0.0053	1/3	PTK2B
progesterone	0.0073	2/10	ESR1, AR
tocilizumab	0	1/1	IL6R

## Data Availability

All the raw data used in this work can be found in the description in the Materials and Methods section. All the processed code is available from the corresponding author upon request.

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
