# Peer review of "Drug-Target Network Study Reveals the Core Target-Protein Interactions of Various COVID-19 Treatments"

_genes, 2022, doi:10.3390/genes13071210_

Round 1

Reviewer 1 Report

The authors constructed a drug-target network called COVID19-DrugNET providing an insight into the interactions between drug targets and proteins. The amount of data analyzed and the number of subjects are admirable. This work is well-designed and deserves the attention of the scientific community.

There are a few factors that would improve the clarity of the paper, listed below:

  • Line #34: missing an “s” at the end of this line

  • Line #227: lack of evidence. Can you provide a supplementary figure to show that the shortest path distance from random sampling is larger than the observed distance of 2.46?

  • Line #237: why there are only 15 points in Figure S3? According to results from line #235, there should be 680 genes within the subnetwork. Can you provide more details about it? Like a table showing genes and their degrees, and a summary table showing the exact number of vertex degrees and vertex frequencies.

Author Response

The authors constructed a drug-target network called COVID19-DrugNET providing an insight into the interactions between drug targets and proteins. The amount of data analyzed and the number of subjects are admirable. This work is well-designed and deserves the attention of the scientific community.

There are a few factors that would improve the clarity of the paper, listed below:

1.Line #34: missing an “s” at the end of this line

Response: We thank the reviewer for the cheering comments and the careful review. We changed the sentence in the abstract to “The lung mast cell was most enriched for target genes among 1,355 human tissue-cell types”.

2.Line #227: lack of evidence. Can you provide a supplementary figure to show that the shortest path distance from random sampling is larger than the observed distance of 2.46?

Response: We thank the reviewer for this suggestion. In this revision, we made a supplementary figure (Figure S3) and provided the statistics of the 100 random sampling experiments of the distance between any two genes of random genes in BioGRID  (min = 2.71;max = 2.81;mean = 2.75;se = 0.02).

3.Line #237: why there are only 15 points in Figure S3? According to results from line #235, there should be 680 genes within the subnetwork. Can you provide more details about it? Like a table showing genes and their degrees, and a summary table showing the exact number of vertex degrees and vertex frequencies.

Response: We thank the reviewer for pointing out this misunderstanding part. In Figure S3, Due to the sparsity of vertices degrees, we binned 680 genes into 15 categories by their log2 (vertices degree values) evenly (X-axis). Then, we calculated the total number of vertices in each binned category (Y-axis in log2 value). We rewrote the caption of Figure S3 accordingly. We added a Table S3 to show the degrees of 680 genes.

Reviewer 2 Report

This is a timely and important work on drug target discovery for COVID-19 treatments. The authors used literature mining and bioinformatic analysis to identify proteins modules associated with COVID-19 medications. The study is well designed and technically sound. A comprehensive bioinformatic analysis, including text mining, network module detection, functional enrichment test, GWAS summary statistics collection and analysis, and differential expression analysis using COVID-19 single-cell RNA-seq data, was conducted. These analyses provide rich information and convincing evidences to support the candidate protein identification. The main contribution of the work is the construction of an actionable human target protein module for developing COVID-19 treatment.

A typo was noticed in the abstract: “The lung mast cell wa 34 most enriched for target genes among 1,355 human tissue-cell types”.

Author Response

Manuscript ID: genes-1795211

Title: Drug-Target Network Study Reveals the Core Target-protein Interactions of Various COVID-19 Treatments

We highly appreciate the valuable comments from reviewers and the editor. During this revision, we further enhanced the manuscript writing including clarity and grammatical error correction. The changed text is MARKED in the revised manuscript. The following is an outline of our point-to-point responses to the editor’s comments.

Reviewer 2:

This is a timely and important work on drug target discovery for COVID-19 treatments. The authors used literature mining and bioinformatic analysis to identify proteins modules associated with COVID-19 medications. The study is well designed and technically sound. A comprehensive bioinformatic analysis, including text mining, network module detection, functional enrichment test, GWAS summary statistics collection and analysis, and differential expression analysis using COVID-19 single-cell RNA-seq data, was conducted. These analyses provide rich information and convincing evidences to support the candidate protein identification. The main contribution of the work is the construction of an actionable human target protein module for developing COVID-19 treatment.

1.A typo was noticed in the abstract: “The lung mast cell wa 34 most enriched for target genes among 1,355 human tissue-cell types”.

Response: We thank the reviewer for the kind comments and the careful review. We changed the sentence in the abstract to “The lung mast cell was most enriched for target genes among 1,355 human tissue-cell types”.
